# MAGIC: Rethinking Dynamic Convolution Design for Medical Image Segmentation

### Shijie Li
shijieli@stu.scu.edu.cn
College of Computer Science, Sichuan University
Chengdu, China

### Yunbin Tu
tuyunbin22@mails.ucas.ac.cn
School of Computer Science and Technology, University of
Chinese Academy of Sciences
Beijing, China

### Qingyuan Xiang
xiangqingyuan@stu.scu.edu.cn
College of Computer Science, Sichuan University
Chengdu, China

### Zheng Li*
lizheng@scu.edu.cn
College of Computer Science, Sichuan University
Chengdu, China

## Abstract

Recently, dynamic convolution shows performance boost for the CNN-related networks in medical image segmentation. The core idea is to replace static convolutional kernel with a linear combination of multiple convolutional kernels, conditioned on input-dependent attention function. However, the existing dynamic convolution design suffers from two limitations: i) The convolutional kernels are weighted by enforcing a single-dimensional attention function upon the input maps, overlooking the synergy in multi-dimensional information. This results in sub-optimal computations of convolution kernels. ii) The linear kernel aggregation is inefficient, restricting the model's capacity to learn more intricate patterns. In this paper, we rethink the dynamic convolution design to address these limitations and propose multi-dimensional aggregation dynamic convolution (MAGIC). Specifically, our MAGIC introduce a dimensional-reciprocal fusion module to capture correlations among input maps across the spatial, channel, and global dimensions simultaneously for computing convolutional kernels. Furthermore, we design kernel recalculation module, which enhances the efficiency of aggregation through learning the interaction between kernels. As a drop-in replacement for regular convolution, our MAGIC can be flexibly integrated into prevalent pure CNN or hybrid CNN-Transformer backbones. The extensive experiments on four benchmarks demonstrate that our MAGIC outperforms regular convolution and existing dynamic convolution. Code is available at: https://github.com/Segment82/MAGIC

## CCS Concepts

• **Computing methodologies → Image segmentation**.

---

*Corresponding author

---

## Keywords

medical image segmentation; multi-dimensional aggregation dynamic convolution; dimensional-reciprocal fusion; kernel recalculation

### ACM Reference Format:

Shijie Li, Yunbin Tu, Qingyuan Xiang, and Zheng Li. 2024. MAGIC: Rethinking Dynamic Convolution Design for Medical Image Segmentation. In *Proceedings of the 32nd ACM International Conference on Multimedia (MM '24), October 28-November 1, 2024, Melbourne, VIC, Australia.* ACM, New York, NY, USA, 10 pages. https://doi.org/10.1145/3664647.3680754

## 1 Introduction

Automatically segmenting various data modalities (such as CT and MRI scan) is one of the most fundamental yet challenging tasks in medical image analysis [25, 52]. On the one hand, accurate segmentation allows healthcare professionals to access reliable morphological data, and assists in making precise and dependable diagnoses [55]. On the other hand, medical images commonly contain the varying scale of organs or lesions, low contrast, and blurred edges [14, 22]. These distractors pose a formidable challenge to segment the organs with intricate organizational structures.

Due to the scale invariance and inductive bias of convolution operation, CNNs are widely adopted to tackle the above challenge. Accordingly, many pure CNN architectures or hybrid CNN-Transformer architectures have been designed for medical image segmentation [6, 15, 20, 29, 33, 38, 51]. For instance, UNet [38] pioneered the use of fully CNN for medical image segmentation, leveraging convolutions to build hierarchical feature representations. In parallel, TransUNet [6] incorporates the strengths of both CNNs and Transformers, which employs the capability of CNNs to capture high-resolution, informative representations in the spatial domain. With these representations, Transformers can further capture their long-range dependencies for achieving better segmentation results.

These CNN-related methods usually opt for the regular convolutions as their backbones. Nevertheless, since the inherent spatial invariance and channel specificity, regular convolutions struggle to adapt to various visual modalities across different spatial locations. Further, the principle of weight sharing impede their effectiveness in extracting features from varying scale targets with blurred edges. In addition, most of them typically introduce extra convolutional layers or enlarge the dimensions of convolutions, such as kernel

size and channel number. Hence, these methods aiming at boosting CNNs capabilities significantly increase computational costs. In short, the above limitations hinder the models' flexibility and diminish their capacity to generalize effectively, especially when dealing with the intricacies of medical imagery.

Recently, Lei *et al.* [28] attempt to introduce the dynamic convolution [9] for medical image segmentation, in order to mitigate the limitations of regular convolutions. The core idea of this method is to replace static convolutional kernel with a linear combination of multiple convolution kernels, according to the input-dependent Squeeze-and-Excitation (SE) [19] attention mechanism. In doing so, the weight coefficients is adaptive for the organs or lesions with intricate organizational structures. This can increase the representation capability without increasing the depth or width of the network. Furthermore, each convolutional kernel only needs to be computed once, thereby reducing extra computational cost in comparison to regular convolutions.

Despite the encouraging progress, there are two major limitations for this method. (1) The convolutional kernels is weighted by enforcing the SE attention function upon the input features. This operation only processes information from a single channel dimension, which emphasizes the critical feature channels about lesion edges or organ textures, as illustrated in Figure 1 (a). However, other dimensional (spatial, global) information within the input is disregarded, where spatial structural information between pixels contain the unique textures and edges of organs/lesions; global information identifies key features across the entire input. (2) The linear combination of multiple convolutional kernels restricts the model's capacity to learn more intricate patterns, because it mainly relies on the additive property of kernels for combination, while overlooking the aggregation of multiple kernels. For the above limitations, we assume that (1) modeling the synergy among three dimensions helps the convolutional kernels understand "what and where the organ/lesion is", thereby enhancing the model's generalization ability; (2) modeling more diverse context information within input feature to compute the convolutional kernels, while increasing the parameter efficient of kernel aggregation.

In this paper, we tackle the above limitations by proposing a novel **M**ulti-dimensional **AG**gregation dynam**I**c **C**onvolution (MAGIC), which learns dimensional-reciprocal convolutional kernels to capture varying scale of organs or lesions, while aggregating multiple kernels to maintain parameter efficiency. As a plug-and-play replacement, MAGIC can seamlessly substitute the regular convolution in pure CNN or hybrid CNN-Transformer backbones for medical image segmentation. Architecture-wise, given feature maps from the backbones, we first design a *Dimensional-Reciprocal Fusion* (DRF) to capture their correlations across the spatial, channel, and global dimensions in parallel, which can attain a comprehensive understanding for them to compute the multiple convolutional kernels. Then, unlike traditional linear combination, we design a *Kernel Recalculation* (KR), which learns the interaction among multiple kernels to generate scalar for each kernel, thus enhancing the overall parameter utilization. Further, the aggregated convolution kernels is used to distill the feature maps of complex organs or lesions in medical images. Finally, these feature maps are fed into the backbones to obtain the accurate segmentation results.

Our contributions can be summarized as follows:

- We make in-depth analysis for the limitations of regular convolution and vanilla dynamic convolution, when using them to extract the complex feature in the medical image (e.g., varying scale of organs or lesions).
- A novel dynamic convolution called MAGIC is designed to learn dimensional-reciprocal convolutional kernels, while enhancing the parameter efficiency of kernel aggregation.
- We demonstrate the efficacy of MAGIC by comparing it with regular convolution and existing dynamic convolution designs on four challenging medical benchmark datasets. Experiments show that MAGIC with a single convolutional kernel produces superior results, and can rival or outperform existing dynamic convolution methods based on multiple kernels. This substantially reduces the number of parameters required, offering an elegant and parameter-efficient design.

## 2 RELATED WORK

### 2.1 Medical Image Segmentation

Convolutional neural networks (CNNs) have been the de-facto standard for the medical image segmentation [17, 34, 37, 40]. UNet [38] pioneered the application of CNN for medical image segmentation task, which introduced a U-shaped fully convolutional network. Inspired by the simplicity and high performance of UNet, numerous variants of U-shaped CNNs have emerged to enhance the segmentation accuracy of models [21, 23, 26, 57]. Benefiting from the robust feature extraction capabilities of convolution, CNNs play an indispensable role in medical image segmentation. Recently, motivated by the success of Transformer architectures [12, 13, 35, 44, 45], some works have attempted to combine CNNs with Transformers [42, 49, 56]. For example, TransFuse [54] introduces a parallel strategy that explores the balance between combining CNNs and Transformers for maximizing the advantages offered by both. Given that CNN can compensate for the limitations of Transformer structures in attending to local information, the hybrid CNN-Transformer networks have powerful feature learning ability.

### 2.2 Dynamic Convolution

Numerous prior studies have demonstrated the efficacy of dynamic convolution in deep neural networks [11, 24, 41]. Brabandere *et al.* [24] introduced dynamic filters in the convolution layer, conditioned on an input. Yang *et al.* [50] and Chen *et al.* [9] replaced the static convolutional kernels with $n$ convolutional kernels, which were weighted using an attention mechanism over the input. Based on this idea, WeightNet [36] employed grouped fully connected layer to generate the convolutional weight directly. However, this vanilla dynamic convolution resulted in an n-fold increase in the number of convolutional parameters. To mitigated this limitation, Li *et al.* [32] proposed a more compact model via matrix decomposition, which learnt a base kernel and a sparse residual to approximate dynamic convolution. ODConv [31] introduced a more generalized form of dynamic convolution, utilizing multidimensional attention to explore different dimensions of the convolutional kernel space for generating kernel weights. Recently, Lei *et al.* [28] introduced DDConv, a integration of deformable convolution and vanilla dynamic convolution, designed to adaptively change the weight

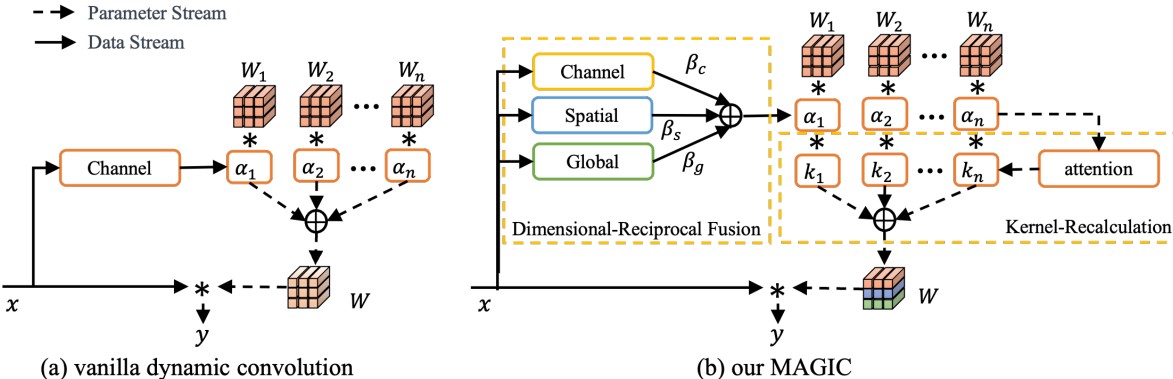

**Figure 1: A comparison of (a) vanilla dynamic convolution and (b) our multi-dimensional aggregation dynamic convolution (MAGIC). Our MAGIC employs a more comprehensive attention mechanism to compute convolutional kernels, and introduces an innovative strategy for kernel aggregation.**

coefficient and deformation offset for increasing the precision of medical image segmentation.

### 2.3 Attention Mechanism

The attention mechanism plays a pivotal role in various vision tasks [8, 19, 43, 46, 53]. For instance, Li *et al.* [30] utilize local information as a guide to spatially activate the feature representation. Hu *et al.* [19] introduce a novel architectural unit called the Squeeze-and-Excitation (SE) block, which computes channel correlation to enhance important channel feature maps. To complement the channel attention block, Chen *et al.* [7] suggest a spatial attention block to refine focus on a feature map. Recently, Wang *et al.* [48] draw inspiration from traditional methods [3] and extend the classical non-local operation to deep neural networks.

### 3 Method

In this section, we first revisit the foundational concepts of vanilla dynamic convolution via a general formulation. Subsequent sections will elaborate on the formulations of our MAGIC.

### 3.1 Revisiting Vanilla Dynamic Convolution

**Definition:** Vanilla dynamic convolution, such as DYConv [9] and CondConv [50], replaces static convolutional kernel with a linear combination of $n$ convolutional kernels $\{W_1, W_2...W_n\}$ weighted by an attention mechanism, as follows:

$$\alpha_i = \pi_i(x) \tag{1}$$

$$y = (\alpha_1 W_1 + \alpha_2 W_2 + ... + \alpha_n W_n) * x \tag{2}$$

where $\{\alpha_1, \alpha_2...\alpha_n\}$ denote attention scalars learned by the attention function $\pi(\cdot)$; $x \in \mathbb{R}^{H \times W \times C}$ and $y \in \mathbb{R}^{H \times W \times C}$ are input and output features, where $H$, $W$, and $C$ denote the height, width, and number of channels respectively. *For brevity, we omit the bias term in this paper.*

**Limitation Discussions.** As described in Equation 2, the dynamic property of vanilla dynamic convolution arises from the aggregation of multiple convolutional kernels, which are computed

based on the exploration of the input space via an attention mechanism $\pi(\cdot)$. Consequently, exploring the input space and aggregating kernels are two key components of dynamic convolution. However, both DYConv [9] and CondConv [50] employ a modified SE [19] structure as the function $\pi(\cdot)$ to capture the channel-wise correlation of the input, thereby generating the attention scalars $\{\alpha_1, \alpha_2...\alpha_n\}$. In other words, these methods overlook the exploration of other dimensional contexts within the input features, such as spatial information and long-range dependencies, which are crucial for understanding the medical dissection of structure and tissue. Furthermore, capturing such relationships is inherently challenging for regular convolution. Such a rudimentary exploration of the input space might be one of the reasons why vanilla dynamic convolution only marginally outperforms traditional convolution in medical image segmentation tasks, as explored in Section 4. Additionally, each convolutional kernel is weighted by attention scalars $\{\alpha_1, \alpha_2...\alpha_n\}$, which are outcomes of the attention mechanism $\pi(\cdot)$. This indicates that the weights of convolutional kernel are exclusively dependent on the input features, treating each kernel with identical importance. As a result, the importance across different kernels are not properly reflected.

### 3.2 Dimensional-Reciprocal Fusion

Building upon the above analysis, we introduce the Dimensional-Reciprocal Fusion (DRF) module to harness synergies from three distinct perspectives: spatial, channel-wise, and global dimensions, as illustrated in Figure 1 (b).

Firstly, we also employ the SE type approach to explore both spatial and channel-wise correlations. Unlike DYConv and Cond-Conv, which focus solely on squeezing the channel-wise dimension, our method compresses both channel and spatial dimensions concurrently. Specifically, for an input maps $x$, we apply average pooling operation along the spatial dimension to obtain the feature maps $\beta_c \in \mathbb{R}^{C \times 1 \times 1}$. Simultaneously, we utilize both average and max pooling operations on the input along the channel dimension. These pooled features are then concatenated, resulting in a feature maps $x' \in \mathbb{R}^{2 \times H \times W}$. Subsequently, we replicate these concatenated

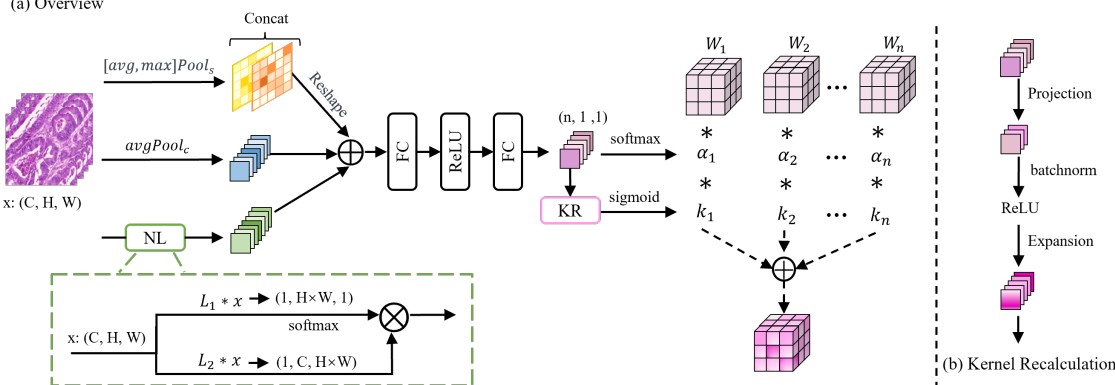

**Figure 2: Our MAGIC comprises two main components: i) Dimensional-Reciprocal Fusion, which models multi-dimensional information to compute convolutional kernels; ii) Kernel Recalculation, which learns the correlation among kernels for aggregation.** $\otimes$ **denotes matrix multiplication;** $\oplus$ **is element-wise addition;** $L_i$ **denotes linear transformation matrices.**

feature maps along the channel dimension $C/2$ times. Finally, we apply average pooling along the spatial dimensions, yielding the condensed feature maps $\beta_s \in \mathbb{R}^{C\times1\times1}$. The process is formulated as:

$$\beta_c = AvgPooling_c(x) \tag{3}$$

$$\beta_s = \mathcal{F}_r * Cat[AvgPooling_s(x), MaxPooling_s(x)] \tag{4}$$

where $Cat[\cdot, \cdot]$ represents the concatenation operation along the channel dimension; $Pooling_{c/s}$ represents the pooling operations along channel-wise and spatial dimensions; $\mathcal{F}_r*$ denotes reshape operation.

Then, to capture unaltered long-range dependencies within an input $x$, we incorporate a non-local operation [3, 48] alongside pooling operations. This operation computes the correlations of a position with all positions in the input map, thereby obtaining the global context. Mathematically, given a input maps $x$, the response $\omega_i$ at position $i$ with all positions in the input maps is defined as:

$$\omega_i = \sum_{j=1}^{N_p} \frac{\mathcal{F}(x_i, x_j)}{C(x)} (L_v * x_j) \tag{5}$$

where $N_p = H \times W$ is the number of positions in the feature map $x$; $L_v*$ is linear transformation matrices; $\mathcal{F}(x_i, x_j)$ denotes the relationship between position i and j; $C(x)$ is a normalization factor. For practical applications, the Embedded Gaussian function, an advanced version of the Gaussian function $\mathcal{F}(x_i, x_j) = e^{x_i^T x_j}$, is utilized to calculate similarity in an embedding space, and $C(x)$ is softmax function:

$$\omega_i = \sum_{j=1}^{N_p} \frac{\exp(< L_q * x_i, L_k * x_j >)}{\sum_{m=1}^{N_p} \exp(< L_q * x_i, L_k * x_m >)} (L_v * x_j) \tag{6}$$

where $L_{k/q}*$ is linear transformation matrices. Inspired by the discoveries in [4] that showed attention maps for different query positions to be nearly identical, we employ a simplified non-local operation which use a query-independent attention map for all

query positions, to further minimize computational demands:

$$\beta_g = \sum_{j=1}^{N_p} \frac{\exp(L_k * x_j)}{\sum_{m=1}^{N_p} \exp(L_k * x_m)} (L_v * x_j) \tag{7}$$

where $\beta_g \in \mathbb{R}^{C\times1\times1}$ is a global attention map. The process as illustrated in Figure 2.

Finally, we combine the feature maps $\{\beta_s, \beta_c, \beta_g\}$, which encapsulate diverse types of information, to compute the attention scalars $\alpha$:

$$\beta = FC(ReLU(FC(\beta_s + \beta_c + \beta_g))) \tag{8}$$

$$\alpha = Softmax(\beta) \tag{9}$$

where $FC(\cdot)$ is fully connected layer. *Essentially, these three types of information are synergistic, providing a robust performance foundation for discerning complex contextual cues. We will validate these advantages with experimental evidence in Section 4.*

### 3.3 Kernel Recalculation

Instead of the traditional linear combination, we introduce Kernel Recalculation (KR) for kernel aggregation to facilitate the interaction among kernels. Specifically, we employ an KR module $\phi(\cdot)$ to calculate correlation scalars $\{k_1, k_2 \ldots k_n\}$ from feature vector $\beta$ in DRF for weighting convolutional kernels:

$$\begin{aligned} W &= (\phi_1(\beta)\alpha_1) W_1 + (\phi_2(\beta)\alpha_2) W_2 + \ldots + (\phi_n(\beta)\alpha_n) W_n \\ &= (k_1\alpha_1)W_1 + (k_2\alpha_2)W_2 + \ldots + (k_n\alpha_n)W_n \end{aligned} \tag{10}$$

where the KR $\phi(\cdot)$ computes the importance of each convolutional kernel. Specifically, it first projects the feature vectors into a compact space, reducing computational costs and identifying essential relationships through a linear matrix mapping. Subsequently, it computes attentions over the number of convolutional kernels. Finally, it generates the weight of kernels by a sigmoid activation function, as shown in Figure 2 (b). The operation of the KR $\phi(\cdot)$ can be formally represented as:

$$\phi_i(\beta) = Sigmoid(L_2 * ReLU(BN(L_1 * \beta))) \tag{11}$$

**Table 1: Results on Synapse multi-organ dataset. DICE scores are reported for individual organs. ↑ denotes higher the better, ↓ denotes lower the better. The best results are in bold.**

| Backbone | Average | | | | Aorta | GB | KL | KR | Liver | PC | SP | SM |
|---|---|---|---|---|---|---|---|---|---|---|---|---|
| | DICE↑ | HD95↓ | mIoU↑ | ASD↓ | | | | | | | | |
| UNet [38] | 76.02 | 33.46 | 65.56 | 6.30 | 86.42 | 59.51 | 78.62 | 71.85 | 92.15 | 59.70 | **86.51** | 73.37 |
| +DYConv [9] (4×) | 77.73 | 30.21 | 67.14 | 4.98 | **88.75** | 62.77 | 83.98 | 75.02 | 92.88 | 60.42 | 85.77 | 71.29 |
| +CondConv [50] (8×) | 77.71 | 33.90 | 67.47 | 5.96 | 87.18 | 65.01 | 83.43 | **78.85** | 92.87 | 59.35 | 87.10 | 68.01 |
| +DCD [32] | 76.87 | 30.29 | 67.84 | 5.97 | 87.52 | 62.42 | 84.14 | 78.65 | 92.83 | 57.06 | 85.67 | 66.63 |
| +ODConv [31] (4×) | 76.56 | 40.73 | 66.67 | 5.12 | 85.51 | 61.11 | 82.13 | 76.45 | 90.43 | 55.10 | 85.48 | 73.27 |
| +DDConv [28] | 77.16 | 28.24 | 65.85 | 5.18 | 87.93 | 64.80 | 80.78 | 77.56 | 94.03 | 53.79 | 85.84 | **73.55** |
| +Our (1×) | 78.31 | 28.65 | 67.92 | 5.78 | 87.79 | 68.46 | 82.95 | 75.09 | 92.74 | 59.87 | 86.32 | 73.24 |
| +Our (2×) | 78.56 | 29.76 | 68.36 | 6.09 | 88.39 | 63.56 | 83.28 | 78.57 | 93.18 | **62.99** | 85.69 | 72.80 |
| +Our (4×) | **78.66** | **24.91** | **68.69** | **4.37** | 87.45 | **70.14** | **84.16** | 78.03 | **93.38** | 57.01 | 86.28 | 72.81 |
| ResNet 18 [18] | 72.94 | 30.42 | 61.92 | 5.28 | 83.68 | 62.60 | 79.41 | 69.19 | 92.50 | 43.96 | 84.40 | 67.99 |
| +DYConv [9] (4×) | 73.45 | 33.42 | 62.27 | 5.66 | 80.14 | 64.57 | 81.24 | 71.11 | 91.96 | 46.70 | 84.18 | 67.72 |
| +CondConv [50] (8×) | 74.02 | 23.91 | 63.41 | 5.55 | 84.07 | 63.03 | 81.68 | 71.94 | **93.52** | 48.68 | 80.42 | 68.82 |
| +DCD [32] | 73.19 | 24.16 | 62.23 | 4.74 | 82.16 | **65.67** | 78.56 | 71.84 | 91.71 | 47.09 | 83.79 | 64.73 |
| +ODConv [31] (4×) | 73.00 | 25.87 | 61.46 | 5.50 | 82.89 | 59.22 | 74.58 | 74.57 | 92.70 | 45.16 | 85.88 | 69.91 |
| +DDConv [28] | 74.33 | 25.88 | 63.49 | 5.08 | 84.93 | 62.94 | 75.57 | 70.31 | 92.16 | **54.06** | 85.86 | 68.81 |
| +Our (1×) | 74.01 | 24.81 | 63.18 | 4.90 | 81.85 | 63.54 | **82.67** | 75.15 | 92.56 | 45.09 | 81.43 | 69.79 |
| +Our (2×) | 74.39 | 22.68 | 63.12 | 4.35 | 82.03 | 64.35 | 79.93 | 71.30 | 92.76 | 46.24 | **86.82** | **71.61** |
| +Our (4×) | **75.29** | **22.38** | **64.45** | **3.51** | **84.96** | 62.94 | 82.36 | 71.16 | 93.07 | 49.89 | 86.50 | 70.90 |
| TransUNet [6] | 77.48 | 31.69 | 64.78 | 8.46 | 87.23 | 63.13 | 81.87 | 77.02 | 94.08 | 55.86 | 85.08 | 75.62 |
| +DYConv [9] (4×) | 79.24 | 31.37 | 68.22 | 5.21 | **88.91** | 67.57 | 82.32 | 79.65 | **95.58** | 60.06 | 87.86 | 72.99 |
| +CondConv [50] (8×) | 78.98 | 27.24 | 67.95 | 5.29 | 87.55 | 67.54 | 85.31 | 78.64 | 95.55 | 58.17 | 89.85 | 71.20 |
| +DCD [32] | 78.18 | 30.82 | 68.88 | 6.51 | 87.20 | 66.13 | 81.83 | 78.90 | 94.35 | 58.32 | 87.57 | 70.12 |
| +ODConv [31] (4×) | 77.76 | 32.98 | 66.91 | 5.41 | 87.88 | 64.06 | 83.00 | 77.02 | 94.25 | 57.16 | 87.28 | 71.77 |
| +DDConv [28] | 78.32 | 22.32 | 68.02 | 4.38 | 87.26 | 67.71 | 82.71 | 77.87 | 93.11 | 63.37 | 86.69 | 67.82 |
| +Our (1×) | **81.42** | **18.84** | **72.29** | **3.11** | 88.27 | **68.18** | 86.09 | **84.44** | 95.27 | **64.74** | **91.18** | 73.23 |
| +Our (2×) | 80.42 | 19.34 | 70.98 | 3.29 | 87.28 | 66.79 | **86.36** | 82.91 | 94.85 | 58.62 | 90.86 | **75.67** |
| +Our (4×) | 79.02 | 28.04 | 69.04 | 4.56 | 88.49 | 66.02 | 83.54 | 77.93 | 94.29 | 59.36 | 89.02 | 73.50 |

where $BN(\cdot)$ represents batch normalization, and $L_{1/2}*$ are the learnable linear matrices. The core of aggregation is that the calculation of the convolutional kernel is no longer oriented towards a single input feature, and the interactions between the kernels also affect the calculation of the convolutional kernel. Moreover, the utilization of a non-linear combination facilitated by the attention function allows for a more nuanced and flexible adaptation to the complexities inherent in feature representation.

## 4 Experiment

### 4.1 Dataset

**Synapse Dataset.** The Synapse multi-organ dataset [27] comprises 30 abdominal CT scans, totaling 3779 axial contrast-enhanced abdominal CT images. Each CT scan consists of 85-198 slices of 512×512 pixels, with a voxel spatial resolution of ([0:54-0:54]×[0:98-0:98]×[2:5-5:0])$mm^3$. Following TransUNet [6], we split the dataset into 18 scans (2211 axial slices) for training, and 12 for validation.

**ACDC Dataset.** The Automatic Cardiac Diagnosis Challenge (ACDC) [2] dataset contains MRI images of 100 patients. The task is to segment the cavity of the right ventricle (RV), the myocardium of the left ventricle (Myo), and the cavity of the left ventricle (LV).

Following [47], we split the dataset into 70 (1304 axial slices), 10 (182 axial slices), and 20 cases for training, validation, and testing.

**GlaS Dataset.** GLAnd segmentation (GlaS) datatset [39] comprises microscopic images of slides stained with Hematoxylin and Eosin (H&E). The dataset includes 165 images: 85 images designated for training purposes and 80 images allocated for testing.

**Skin Lesion Segmentation.** We utilize the ISIC 2017 dataset [10] for skin lesion segmentation, consisting of 2000 dermoscopic images for training, 150 for validation, and 600 for testing. Following the setting in [1], we resize all images to 192×256.

### 4.2 Implementation Details

**Backbones.** We employ UNet [38] and ResNet [18] as CNNs backbone to assess dynamic convolution. Following DYConv [9] and CondConv [50], we apply dynamic convolution for all convolution layers except the first layer. Besides, we use TransUNet [6] and TransFuse [54] as hybrid CNN-Transformer backbones for a comprehensive evaluation.

**Experimental Setup.** We utilize UNet, ResNet, and TransUNet architectures with dynamic convolutions on the Synapse and ACDC datasets. For the GlaS dataset, we apply both TransUNet and UNet

**Table 2: Results on the ACDC dataset. DICE scores are reported for individual organs. The best results are in bold.**

| Backbone | mDice | HD95 | RV | Myo | LV |
|---|---|---|---|---|---|
| UNet | 88.89 | 3.31 | 84.70 | 86.97 | 95.02 |
| +DYConv (4×) | 89.48 | 4.40 | 86.30 | 87.00 | 95.13 |
| +CondConv (8×) | 90.03 | 3.98 | 86.86 | 88.04 | 95.19 |
| +DCD | 88.98 | 3.92 | 85.80 | 86.88 | 94.26 |
| +ODConv (4×) | 89.36 | 1.95 | 85.68 | 87.28 | 95.13 |
| +DDConv | 90.82 | 2.16 | 87.24 | 89.34 | 95.88 |
| +Our (1×) | 90.42 | 2.96 | 87.73 | 88.07 | 95.47 |
| +Our (2×) | **91.87** | **1.92** | **89.69** | **90.01** | **95.90** |
| +Our (4×) | 91.67 | 2.18 | 89.19 | 89.83 | 95.83 |
| ResNet 18 | 87.19 | 3.26 | 83.56 | 84.25 | 93.78 |
| +DYConv (4×) | 88.39 | 2.26 | 85.41 | 85.56 | 94.20 |
| +CondConv (8×) | 88.45 | 1.98 | 85.46 | 85.76 | 94.13 |
| +DCD | 87.78 | 2.72 | 83.98 | 85.33 | 94.03 |
| +ODConv (4×) | 88.08 | 2.32 | 84.71 | 85.72 | 94.12 |
| +DDConv | 88.43 | 2.37 | 85.39 | 85.61 | 94.32 |
| +Our (1×) | 89.05 | 1.67 | 86.60 | 85.97 | 94.58 |
| +Our (2×) | 90.34 | 1.43 | 88.12 | 87.86 | 95.06 |
| +Our (4×) | **90.54** | **1.20** | **88.47** | **87.91** | **95.27** |
| TransUNet | 89.71 | 3.01 | 88.86 | 84.53 | 95.73 |
| +DYConv (4×) | 90.54 | 1.30 | 87.23 | 88.83 | 95.55 |
| +CondConv (8×) | 90.49 | 1.95 | 87.80 | 88.17 | 95.51 |
| +DCD | 90.74 | 2.03 | 87.89 | 88.78 | 95.56 |
| +ODConv (4×) | 90.18 | 1.25 | 88.58 | 87.22 | 94.76 |
| +DDConv[28] | 90.48 | 1.28 | 88.35 | 87.82 | 95.26 |
| +Our (1×) | **91.76** | 1.15 | **89.64** | 89.45 | 95.73 |
| +Our (2×) | 91.25 | 1.26 | 88.61 | 89.24 | **95.91** |
| +Our (4×) | 91.63 | **1.08** | 89.57 | **89.47** | 95.87 |

**Table 3: Results on the GlaS dataset. The best results are in bold.**

| Backbone | mDice | mIoU | HD95 |
|---|---|---|---|
| UNet | 84.48 | 74.02 | 28.69 |
| +DYConv (4×) | 85.08 | 74.97 | 25.40 |
| +CondConv (8×) | 84.86 | 74.42 | 27.00 |
| +DCD | 84.60 | 72.30 | 30.92 |
| +ODConv (4×) | 85.00 | 75.49 | 25.63 |
| +DDConv | 85.46 | 75.01 | 25.34 |
| +Our (1×) | 85.26 | 75.10 | 25.43 |
| +Our (2×) | 86.02 | 74.48 | 24.32 |
| +Our (4×) | **86.71** | **76.11** | **22.54** |
| Transunet | 86.34 | 77.34 | 24.68 |
| +DYConv (4×) | 86.99 | 78.40 | 26.61 |
| +CondConv (8×) | 86.45 | 77.71 | 23.25 |
| +DCD | 85.71 | 76.31 | 29.44 |
| +ODConv (4×) | 86.64 | 77.78 | **22.24** |
| +DDConv | 85.37 | 75.94 | 26.72 |
| +Our (1×) | 86.72 | 78.00 | 23.07 |
| +Our (2×) | **87.67** | **79.20** | 24.95 |
| +Our (4×) | 87.27 | 78.37 | 25.17 |

**Table 4: Results on the ISIC 2017 skin lesion segmentation benchmarks. The best results are in bold.**

| Backbone | mDice | mIoU | ACC |
|---|---|---|---|
| TransFuse [54] | 81.52 | 72.31 | 91.92 |
| +DYConv (4×) | 81.67 | 72.40 | 92.81 |
| +CondConv (8×) | 82.74 | 73.69 | 92.86 |
| +DCD | 81.08 | 71.42 | 91.10 |
| +ODConv (4×) | 81.15 | 72.43 | 92.12 |
| +DDConv | 83.05 | 74.40 | 92.49 |
| +Our (1×) | 82.95 | 74.99 | 92.35 |
| +Our (2×) | 83.11 | 74.44 | 92.72 |
| +Our (4×) | **84.02** | **75.50** | **92.92** |

for assessments. Additionally, TransFuse with dynamic convolution is evaluated on the ISIC 2017 dataset. Within the ResNet framework, we integrate a lightweight Hamburger module [16] as the segmentation head. For the UNet and TransUNet frameworks, dynamic convolution is implemented in the CNN encoder. All models were trained for 150 epochs on the Synapse dataset. For the ACDC dataset, training was extended to 200 epochs. Following the TransFuse, all models were trained for 30 epochs on the ISIC 2017 dataset. For the GlaS dataset, the models were trained for 100 epochs. For fair-comparison, we use publicly available codes and adhere to popular training and testing configurations used by the community. All models are trained under same settings without any pre-trained models.

### 4.3 Evaluation Metrics

In the Synapse dataset, we employ mean Dice coefficient (mDICE), Hausdorff Distance at 95th percentile (HD95), mean Intersection over Union (mIoU), and Average Surface Distance (ASD) as evaluation metrics, reporting DICE scores for individual organs. We utilize mDICE and HD95 for evaluating the ACDC dataset. In the GlaS dataset, we incorporate mDICE, mIoU and HD95 as the metrics. For the ISIC 2017 dataset, we employ four metrics for assessment: mDice, Sensitivity (SE), Accuracy (ACC), and Specificity (SP).

### 4.4 Comparative Results

**Comparative Results on Synapse Dataset.** Table 1 presents a comparison of our MAGIC against existing dynamic convolution methods in three backbone architectures (UNet, ResNet 18, TransUNet) on the Synapse dataset. We can observe that our MAGIC always outperforms other methods, achieving the highest performance gains on all backbones. In contrast, other dynamic convolutions, such as ODConv (4×), provides slight enhancements in the mDice score. Furthermore, backbone models with our MAGIC significantly reduce the HD95 score, indicating their ability to capture finer structures and generate more precise contours. This improvement is attributed to the proposed attention function, which effectively harnesses the multi-dimensional information within the input space for computing kernels.

**Comparative Results on ACDC Dataset.** From the results shown in Table 2, we can observe similar performance improvement trends as on the Synapse dataset. For UNet and TransUNet

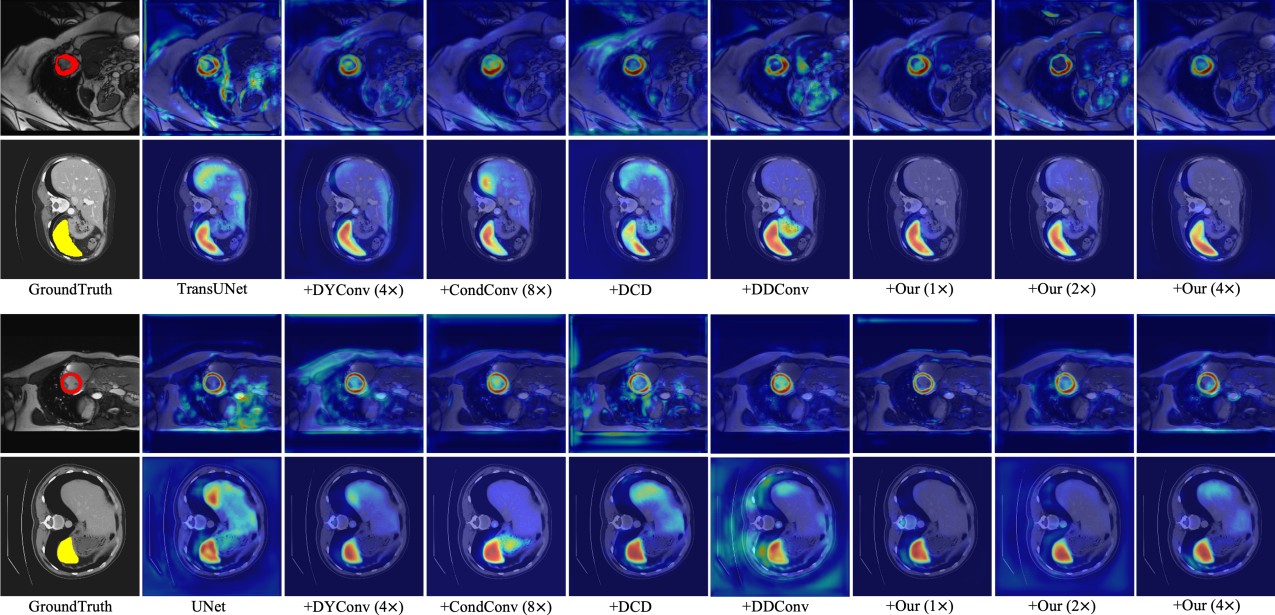

**Figure 3: Visualization of class activation maps in different structures with Grad-CAM++ [5]. Result are obtained from the TransUNet [6] and UNet [38] with different dynamic convolution on the ACDC (above) and Synapse (below) datasets.**

**Table 5: Comparison of performance and computation on the Synapse dataset. FLOPs are calculated using an input size of 256×256.**

| Backbone | Params | GFLOPs | DICE |
|---|---|---|---|
| ResNet 18 | 12.34 | 9.671 | 72.94 |
| +DYConv (4×) | 45.60 | 9.676 | 73.45 |
| +CondConv (8×) | 89.29 | 9.676 | 74.02 |
| +DCD | 15.86 | 9.763 | 73.19 |
| +ODConv (4×) | 45.55 | 9.724 | 73.00 |
| +DDConv | 93.69 | 10.14 | 74.33 |
| +Our (1×) | 17.03 | 9.674 | 74.01 |
| +Our (2×) | 28.02 | 9.679 | 74.39 |
| +Our (4×) | 49.99 | 9.685 | 75.29 |

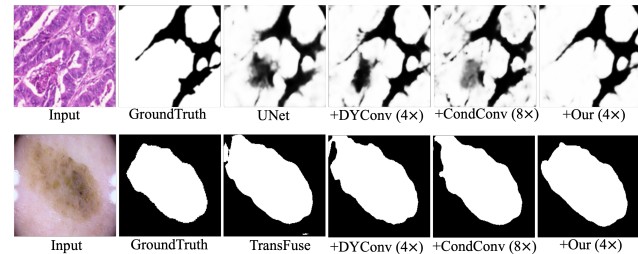

**Figure 4: Visualization of segmentation from our MAGIC and other dynamic convolution on GlaS and ISIC 2017 datasets.**

backbones, DYConv (4×) and CondConv (8×) yield mDICE improvements of 0.59%/0.83% and 1.14%/0.78% over baseline models, respectively. Our method, MAGIC (1×), utilizing a single convolutional kernel, significantly outperforms these, with mDICE improvements of 1.53%/2.05%.

**Comparative Results on GlaS Dataset.** In experiments conducted on the GlaS dataset, our MAGIC also outperforms other dynamic convolutions, as demonstrated in Table 3. We can see that UNet with our MAGIC (4×) significantly enhances the segmentation accuracy, whereas the use of other dynamic convolutions results in minimal improvement.

**Comparative Results on Skin Lesion Segmentation Dataset.** The comparison results for the ISIC 2017 skin lesion segmentation benchmarks against the existing dynamic convolutions are presented in Table 4. The experimental results demonstrate that our

MAGIC achieves the highest mDICE (84.02%), SE (75.50%), and ACC (92.92%), surpassing TransFuse baseline by 2.50%, 3.19%, and 1.00%, respectively. It's noteworthy that ODConv, despite design an attention mechanism to capture multi-scale kernel space information, falls short by overlooking the synergy between dependencies within the input space, resulting in sub-optimal performance.

**Analysis of Efficiency.** We conduct a quantitative analysis of different dynamic convolution parameters and computational complexity (measured in GFLOPs) using the ResNet 18 backbone, as shown in Table 5. Specifically, our MAGIC notably improves the expressive power of regular convolution with negligible extra computation cost. Compared to other dynamic convolution methods, our approach stands out for its competitiveness in both parameter quantity and computational cost. Note that our MAGIC (4×) outperforms DDConv (75.29% vs. 74.33%), using only about half as many parameters (49.99M vs. 93.69M). *This demonstrates that our MAGIC*

**Table 6: Ablation study of Dimensional-Reciprocal Fusion (DRF) is conducted on the Synapse, ACDC, and GlaS datasets, wherein average DICE scores are reported for each dataset. Investigating the complementarity of modeling spatial information (SP), channel information (CH), and non-local information (NL) for computing convolutional kernels. The best result is bolded.**

| Index | SP | CH | NL | UNet + Our (4×) | | | | TransUNet + Our (4×) | | | |
|-------|----|----|-----|--------|---------|-------|-------|--------|---------|-------|-------|
| | | | | GFLOPs | Synapse | ACDC | GlaS | GFLOPs | Synapse | ACDC | GlaS |
| A. 1 | × | × | × | 54.675 | 76.53 | 89.07 | 84.69 | 38.517 | 77.79 | 90.16 | 86.84 |
| A. 2 | ✓ | × | × | 54.681 | 77.22 | 88.83 | 85.50 | 38.517 | 77.95 | 90.82 | 84.58 |
| A. 3 | × | ✓ | × | 54.676 | 77.92 | 89.32 | 84.23 | 38.519 | 78.18 | 91.01 | 86.98 |
| A. 4 | × | × | ✓ | 54.715 | 77.68 | 89.45 | 85.54 | 38.527 | 78.37 | 90.78 | 84.71 |
| A. 5 | ✓ | ✓ | × | 54.681 | 78.13 | 90.06 | 85.92 | 38.519 | 78.52 | 91.23 | 86.67 |
| A. 6 | × | ✓ | ✓ | 54.715 | 78.09 | 89.67 | 86.07 | 38.527 | 76.34 | 91.11 | 85.49 |
| A. 7 | ✓ | × | ✓ | 54.720 | 78.39 | 90.14 | 85.81 | 38.529 | 78.33 | 90.85 | 87.01 |
| A. 8 | ✓ | ✓ | ✓ | 54.720 | **78.66** | **90.23** | **86.47** | 38.528 | **79.02** | **91.63** | **87.27** |

**Table 7: Ablation study of the Kernel Recalculation (KR) on the ACDC and GlaS datasets, wherein average DICE scores are reported for each datasets.**

| Index | Backbone | ACDC | GlaS |
|-------|----------|------|------|
| B. 1 | UNet+Our(2×) | 91.87 | 86.02 |
| B. 2 | +w/o KR | 91.25 | 85.50 |
| B. 3 | UNet+Our(4×) | 91.67 | 86.71 |
| B. 4 | +w/o KR | 91.13 | 85.65 |
| B. 5 | TransUNet+Our(2×) | 91.25 | 87.67 |
| B. 6 | +w/o KR | 90.83 | 86.53 |
| B. 7 | TransUNet+Our(4×) | 91.63 | 87.27 |
| B. 8 | +w/o KR | 90.95 | 86.34 |

*achieves the best trade-off between segmentation performance and computational complexity.*

**Visualization Results.** Figure 3 illustrates the class activation maps generated by UNet/TransUNet employing different dynamic convolutions on the ACDC and Synapse datasets. Our MAGIC, in contrast to other dynamic convolutions, showcases a more precise focus on the segmentation area and provides a finer delineation of the segmentation region. Impressively, we achieve enhanced segmentation quality using only a single convolutional kernel. Furthermore, we present segmentation results of baseline models (UNet and TransFuse) with our MAGIC and other dynamic convolutions on the GlaS and ISIC 2017 datasets, as shown in Figure 4. Our MAGIC significantly reduces the number of false positive segments compared to other dynamic convolutions. Particularly in segmenting complex targets on the GlaS dataset, our results align more closely with the ground truth, leading to fewer inaccurately segmented areas.

### 4.5 Ablation Study

**Effectiveness of Dimensional-Reciprocal Fusion.** To assess the effectiveness of Dimensional-Reciprocal Fusion (DRF), we integrate MAGIC (4×) into the UNet and TransUNet architectures. Table 6 clearly demonstrates the synergistic effect of incorporating different dimensional information for computing kernels. Compared

to UNet/TransUNet baselines, Experiment A. 1 shows marginal improvements in mDice scores by 0.51%/0.31%, 0.18%/0.45%, and 0.21%/0.5% on the Synapse, ACDC, and GlaS datasets, respectively. These results highlight the significance of the attention mechanism in dynamic convolution. Analyses of Experiments A. 2, A. 3, and A. 5 reveal that incorporating information from both the channel and spatial dimensions significantly enhances segmentation accuracy. Furthermore, our observations indicate that capturing the global information of the input is essential for enhancing segmentation performance. *Therefore, our study validates the assumption that incorporating three distinct dimensional information into kernel computation yields synergistic advantages.*

**Effectiveness of Kernel Recalculation.** Table 7 presents the effectiveness of Kernel Recalculation (KR), which learns the correlations among kernels to enhance the aggregation. The outcomes demonstrate that our KR surpasses the vanilla linear combination in four experiments, achieving enhancements in the mDice score ranging from 0.43% to 1.25%. *This confirms the assumption that improving the aggregation process aids in comprehending complex patterns within the input features.*

**Effectiveness of Convolutional Kernels Number.** Tables 1, 2, 3, and 4 show the effectiveness of the number of convolutional kernels on segmentation accuracy. A clear trend emerges, showing that increasing the number of convolutional kernels results in more precise segmentation of the target. This effect is especially notable in purely convolutional networks. Moreover, our MAGIC showcases remarkable adaptability with Transformer architectures, sustaining strong performance despite the use of a minimal set of convolutional kernels (such as, one or two). *This affirms that our MAGIC approach can be used to replace regular convolutions in many architectures.*

## 5 Conclusion

In this paper, we propose MAGIC, an elegant form of dynamic convolution for medical image segmentation. The key insight is to utilize the rich information within the input for computing multiple convolutional kernels. To further enhance segmentation performance, we employ a non-linear aggregation strategy that effectively leverages the power of multiple kernels. Extensive experiments demonstrate that our MAGIC outperforms existing dynamic convolution methods on four popular medical datasets considerably.

## Acknowledgments

This work was supported in part by the National Key Research and Development Program of China under Grant 2020YFA0714003, in part by the Science and Technology Planning Project of Sichuan Province under Grant 2021YFQ0059, and in part by the National Natural Science Foundation of China under Grant No. 61471250.

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
