# OpenReview forum: "MAGIC: Rethinking Dynamic Convolution Design for Medical Image Segmentation"
_acmmm.org/ACMMM/2024/Conference — MM2024 Poster_

### Official Review · Reviewer_5kc1 · 2024-05-24

**Rating:** 5
**Confidence:** 4

**Summary:**

This article proposes a dynamic convolution kernel named MAGIC, primarily utilizing DRF and KR modules to capture spatial, channel, and global information, while also considering the importance of the convolution kernel itself. Extensive experiments were conducted on numerous datasets.

**Strengths:**

Excellent innovations, good illustrations, ample comparative experiments, and mature writing skills make me believe this is an outstanding paper.

**Limitations:**

- In the abstract, the authors mention that a drawback of dynamic convolution is "The linear kernel aggregation is inefficient." However, the authors' KR method merely takes into account the importance of the convolution kernel itself, and ultimately also performs linear aggregation.

- In Figure 2, I believe the input map processed through [avg, max] pooling results in a shape of [2, H, W], rather than [C, 1, 1] as mentioned in line 343. How is [2, H, W] reshaped to [C, 1, 1] and added to the other attention-extracted tensors?

- In line 453, L1/L2 are linear matrices. Are they learnable, or are they consistent with L1/L2 in the Non-Local section? I believe the authors need to provide a clearer description of this part.

- Why are different networks used for different datasets? ACDC employs three backbone networks, whereas ISIC uses only one backbone network. To me, such an experimental setup is somewhat insufficient and fails to fully convince me.

- In Table 6, it is somewhat unusual to place upward arrows next to the dataset names. Typically, upward and downward arrows should be placed next to the metrics. I believe the authors could revise the table accordingly. Similar issues appear in other tables as well.

- In line 847, does "using only a single convolutional kernel" mean (1×)? Moreover, the paper seems to lack an explanation of the meanings of 1×, 2×, and 4×. I suggest adding some basic descriptions.

- In line 902, the section "Effectiveness of Convolutional Kernels Number" mentions that in previous comparative experiments, the performance of 1× or 2× convolutional kernels was sometimes better than that of 4× convolutional kernels. This is somewhat counterintuitive and does not align with the authors' explanations. I suggest adding text or experiments to investigate and discuss this phenomenon.

- Including inference time metrics in Table 5 would be better.

**Suitability:**

2

---

### Official Review · Reviewer_cycV · 2024-05-24

**Rating:** 5
**Confidence:** 4

**Summary:**

The paper proposes a novel dynamic convolution method called "MAGIC" for medical image segmentation tasks. The key contributions are:

Introducing a dimensional-reciprocal fusion module to capture correlations among input maps across spatial, channel, and global dimensions for computing convolutional kernels.

Designing a kernel recalculation module to enhance the efficiency of kernel aggregation by learning the interaction between kernels.

Demonstrating the effectiveness of MAGIC as a drop-in replacement for regular convolution in both pure CNN and hybrid CNN-Transformer backbones.

**Strengths:**

The proposed MAGIC method addresses two key limitations of existing dynamic convolution approaches - sub-optimal computation of convolution kernels and inefficient linear kernel aggregation.

The dimensional-reciprocal fusion and kernel recalculation modules are novel and well-motivated to improve the flexibility and representation capacity of dynamic convolutions.

Extensive experiments on four medical image segmentation benchmarks show MAGIC outperforms regular convolution and existing dynamic convolution methods.

The paper is well-structured, with a clear explanation of the motivations, method details, and empirical evaluation.

**Limitations:**

1. More detailed implementation specifics would be greatly beneficial to the readers. These specifics could include hyperparameter settings, training procedures, and ablation studies. The ablation studies, in particular, would help to understand the individual contributions of the proposed components within your method.
2. It would be helpful to compare your method with a wider range of state-of-the-art medical image segmentation methods. Currently, the comparison seems to be limited to just regular convolution and existing dynamic convolution methods. A more comprehensive evaluation would provide a broader context for the effectiveness of your method.
3. Discussing potential limitations or failure cases of the MAGIC method would give readers a more balanced perspective. It would be beneficial to know what circumstances might limit the effectiveness of the proposed method.
4. Can you provide more insights into how the dimensional-reciprocal fusion module captures the correlations across spatial, channel, and global dimensions? Understanding the key design choices and intuitions behind this module could help others to further refine or build upon your work.
5. How does the kernel recalculation module enhance the efficiency of kernel aggregation? Can you elaborate on the specific advantages it provides over a simple linear aggregation? This could provide more clarity on the unique benefits offered by your method.
6. What are the computational and memory cost differences between MAGIC and the regular convolution or existing dynamic convolution methods? A quantitative comparison would be helpful. This would provide a clear picture of the trade-offs involved in using your method over other existing ones.
7. Some significant literature could be further included to improve the comprehensiveness of the literature review regarding medical image segmentation and efficient network designs. The current innovation is limited due to the lack of a more detailed literature review of the precedent.

Hu S, Liao Z, Zhang J, et al. Domain and content adaptive convolution based multi-source domain generalization for medical image segmentation[J]. IEEE Transactions on Medical Imaging, 2022, 42(1): 233-244.
Li C, Lin M, Ding Z, et al. Knowledge condensation distillation[C]//European Conference on Computer
Vision. Cham: Springer Nature Switzerland, 2022: 19-35.
Sun X, Chen C, Wang X, et al. Gaussian dynamic convolution for efficient single-image segmentation[J]. IEEE Transactions on Circuits and Systems for Video Technology, 2021, 32(5): 2937-2948.
Li C, Ma W, Sun L, et al. Hierarchical deep network with uncertainty-aware semi-supervised learning for vessel segmentation[J]. Neural Computing and Applications, 2022: 1-14.
Sun L, Li C, Ding X, et al. Few-shot medical image segmentation using a global correlation network with discriminative embedding[J]. Computers in biology and medicine, 2022, 140: 105067.
Li C, Zhang Y, Liang Z, et al. Consistent posterior distributions under vessel-mixing: a regularization for cross-domain retinal artery/vein classification[C]//2021 IEEE International Conference on Image Processing (ICIP). IEEE, 2021: 61-65.
Lei T, Zhang D, Du X, et al. Semi-supervised medical image segmentation using adversarial consistency learning and dynamic convolution network[J]. IEEE Transactions on Medical Imaging, 2022.

**Suitability:**

2

---

### Official Review · Reviewer_RsVS · 2024-05-24

**Rating:** 3
**Confidence:** 2

**Summary:**

This paper enhances dynamic convolution through a dimensional-reciprocal fusion module and a kernel recalculation module. Unlike previous works [11, 45] that use the SE module [21], the proposed approach integrates spatial, channel, and non-local information. The kernel recalculation module then assesses the importance of these kernels. Experimental results validate the method's effectiveness.

**Strengths:**

- The paper is well-organized.
- The experiments are comprehensive, and the ablation study highlights the significance of each proposed module.

**Limitations:**

- The design of the kernel recalculation appears to duplicate the attention function. Both \( k \) and \( \alpha \) are generated based on the fusion feature \( \beta \) through a nonlinear operation. Since \( \beta \) already fuses the feature map from multiple channels [11], it contradicts the motivation to introduce \( \alpha \), which seems to have a similar function.
- The meaning of '4x' and '8x' is unclear. The author should explain this notation.
- The Synapse dataset [29] includes 13 organs, but Table 1 only shows results for some of them. Please clarify whether the experiments were conducted on all organs or only a subset.
- There are some typos. In Row 341, the shape of \( \beta_c \) should be \( h \times w \).

**Suitability:**

2

---

### Official Review · Reviewer_RpMy · 2024-05-25

**Rating:** 3
**Confidence:** 3

**Summary:**

The paper “MAGIC: Rethinking Dynamic Convolution Design for Medical Image Segmentation” introduces a new form of dynamic convolution called MAGIC, which aims to address the problems of computing multiple convolutional kernels on dynamic convolution. Specifically, dimensional-reciprocal fusion module is proposed to capture correlations among input maps. Furthermore, MAGIC contains a kernel recalculation module to enhance the efficiency of aggregation by learning the interaction between kernels. Besides, this paper aiming at the challenge current dynamic convolution faced, logically and rigorously explaining the method MAGIC.

**Strengths:**

S1: The introduction of MAGIC presents an effective approach to consider the impact of other dimensional contexts (spatial information and long-range dependencies) on dynamic convolution, the importance across different kernels is taken into account.

S2: This paper demonstrates the issues that occurred in the previous work, based on these issues, a well-structured illustration of the two ideas (Dimensional-Reciprocal Fusion, Kernel Recalculation) proposed in this paper.

**Limitations:**

W1: The experimental data in this paper is not sufficient to prove validity. As this paper is about medical image segmentation, the CiT-Net[*] conducted experiments on dataset ISIC2018, but this paper is on ISIC2017, and the results of CiT-Net is better. The reviewer considers the results may not clearly demonstrate the method in this paper is more effective for medical image segmentation, experiments based on CiT-Net backbone are required.

W2: Doubts about the purpose of the MAGIC raised by this paper. The title contains “for Medical Image Segmentation”, but this paper is mainly to deal with the dynamic convolution, in the third part, it is not seen that MAGIC is specifically targeted at medical image segmentation. The reviewer is curious about the effect of this method in other downstream tasks, such as object detection, image classification.

W3: The grammar of the paper needs to be carefully checked, there are some article errors in the paper.

[*] Lei T, Sun R, Wang X, et al. CiT-Net: convolutional neural networks hand in hand with vision transformers for medical image segmentation[J]. arXiv preprint arXiv:2306.03373, 2023.

**Suitability:**

2

---

### Meta-Review · Area_Chair_oxhe · 2024-06-30

**Recommendation:** Accept (Poster)
**Confidence:** 4

**Metareview:**

The reviewers agreed that the paper is reasonably novel and the experiments are reasonably convincing for publication. They also agreed that many parts of the paper should be revised to make the novelty and motivation clear, to include more references, and to better explain the method.